# Multi-Responsive Sensor Based on Porous Hydrogen-Bonded Organic Frameworks for Selective Sensing of Ions and Dopamine Molecules

**DOI:** 10.3390/molecules27248750

**Published:** 2022-12-09

**Authors:** Faqiang Chen, Hui Xu, Youlie Cai, Wei Zhang, Penglei Shen, Wenhua Zhang, Hangqing Xie, Gongxun Bai, Shiqing Xu, Junkuo Gao

**Affiliations:** 1Key Laboratory of Rare Earth Optoelectronic Materials and Devices of Zhejiang Province, Institute of Optoelectronic Materials and Devices, Collage of Optical and Electronic Technology, China Jiliang University, Hangzhou 310018, China; 2Institute of Functional Porous Materials, The Key Laboratory of Advanced Textile Materials and Manufacturing Technology of Ministry of Education, College of Materials and Textiles, Zhejiang Sci-Tech University, Hangzhou 310018, China; 3Technical Center of Hangzhou Customs, Hangzhou 310016, China

**Keywords:** hydrogen-bonded organic frameworks (HOFs), excimer formation, fluorescence sensor, ions sensing, dopamine sensing

## Abstract

Hydrogen-bonded organic frameworks (HOFs), as an emerging porous material, have attracted increasing research interest in fluorescence sensing due to their inherent fluorescence emission units with unique physicochemical properties. Herein, based on the organic building block 3,3′,5,5′-tetrakis-(4-carboxyphenyl)-1,1′-biphenyl (H_4_TCBP), the porous material HOF-TCBP was successfully synthesized using hydrogen bond self-assembly in a DMF solution. The fluorescence properties of the HOF-TCBP solution showed that when the concentration was high, excimers were easily formed, the PL emission was red-shifted, and the fluorescence intensity became weaker. HOF-TCBP showed good sensitivity and selectivity to metal ions Fe^3+^, Cr^3+^, and anion Cr_2_O_7_^2−^. In addition, HOF-TCBP can serve as a label-free fluorescent sensor material for the sensitive and selective detection of dopamine (DA). HOF-based DA sensing is actually easy, low-cost, simple to operate, and highly selective for many potential interfering substances, and it has been successfully applied to the detection of DA in biological samples with satisfactory recoveries (101.1–104.9%). To our knowledge, this is the first report of HOF materials for efficient detection of the neurotransmitter dopamine in biological fluids. In short, this work widely broadens the application of HOF materials as fluorescent sensors for the sensing of ions and biological disease markers.

## 1. Introduction

Hydrogen-bonded organic frameworks (HOFs) are a class of crystalline porous materials self-assembled by organic molecules units through hydrogen bonding interactions [1,2,3,4,5]. In addition to hydrogen bonding interactions, there are other non-covalent interactions, such as electrostatic interactions, π–π stacking, and other intermolecular forces such as van der Waals forces, which also play an important role in the construction and stability of HOFs [6,7,8]. The rational construction of HOFs with permanent porosity and the utilization of large π-conjugated system aromatics with highly rigid organic molecules as the building blocks of HOFs usually exhibit excellent fluorescence properties, making them promising luminescent materials [9,10,11]. Due to the weak, flexible, reversible, and directional force of hydrogen bonding, HOFs have excellent solid solution processing properties, structural flexibility, and easy recrystallization and recovery [12,13,14]. In addition, this material has the advantages of large specific surface area, mild synthesis conditions, good biocompatibility, and low cytotoxicity [15,16,17,18,19,20]. The porosity and large specific surface area of HOFs materials help to fully contact the analyte, facilitate interactions with guest molecules, thereby improving detection sensitivity. The differential recognition of binding sites makes it highly selective. Therefore, it has a good prospect in luminescence sensing. Several pioneering applications have been explored in the past decade, including gas storage and separation [16,21,22,23], chiral separation and structure determination [7], heterogeneous catalysis [24,25], sensing [26,27], and proton conduction [28,29]. Although the detection of metal ions, anions in nature, and disease markers in biological systems has attracted more and more attention and research by scientists [30,31,32], there are few related reports on HOFs-based fluorescence sensors.

One of the indispensable elements in the human body is chromium. Chromium plays a very important role in many areas of industrial production [33,34,35]. For Cr(III) cation, it is an important and essential biological element and is widely used in practical industrial production. However, excess Cr(III) ions will combine with DNA to generate genetic mutations [36]. Iron is an important transition metal element and is present in all aspects of life. At the same time, it plays a vital role in various aspects, such as biology and the environment. Most importantly, iron is the most abundant as well as versatile transition metal in human cellular systems and is involved in a variety of proteins and enzymes that play different important roles, such as oxygen metabolism, electron transport, oxygen uptake, and transcriptional regulation [37]. Iron concentration is an important aspect of human health, as lower or higher iron concentrations may lead to endotoxemia, gastrointestinal disturbances, and decreased immunity. Excessive Fe^3+^ in drinking water may cause many problems related to human health. It is suggested that the indicator of Fe^3+^ in water is 0.3 ppm, so the selective detection of Fe^3+^ is of great significance to human health and the environment [38].

Dopamine (DA) is a catecholamine neurotransmitter that plays an important role in the central nervous system and cardiovascular system [39,40]. It controls many biological functions such as emotion, motivation, motor control, cognition, and endocrine regulation [41,42]. Abnormal concentrations of DA in organisms can lead to various neurological diseases, such as Alzheimer’s disease and Parkinson’s disease. Considering that compared with traditional detection methods, such as chemiluminescence [43], electrochemical [44], high performance liquid chromatography [45], and capillary electrophoresis [46], the use of a fluorescence method has the advantages of high sensitivity, low cost, high practicability, easy operation, and selection.

In this work, we report a porous material HOF-TCBP, which is composed of organic building unit 3,3′,5,5′-tetrakis-(4-carboxyphenyl)-1,1′-biphenyl (H_4_TCBP) and connected by hydrogen bond. HOF-TCBP exhibits good sensitivity and selectivity towards metal ions Fe^3+^, Cr^3+^ and anion Cr_2_O_7_^2−^. Moreover, HOF-TCBP also functions as a label-free fluorescence-based sensor material for the sensitive and selective detection of dopamine. The sensing of DA based on HOF is practically facile, low-cost, and shows adequate performance for biological samples. To the best of our knowledge, there have been very few HOF-based sensors achieve the sensing of metal ions and anions, and this is the first report of HOF material for highly efficient sensing of neurotransmitter dopamine in biological fluids.

## 2. Results and Discussion

The porous HOF material HOF-TCBP was prepared according to previous reports [13]. Synthesis of HOF-TCBP by organic building molecule H_4_TCBP by self-assembly in DMF solution. The porous HOF-TCBP material crystallizes in the orthorhombic Fddd space group with a 5-fold interpenetrating three-dimensional (3D) framework structure. In the crystal structure, due to the rotation of the C-C bond between the two benzene rings in the organic building unit H_4_TCBP biphenyl, the two benzene rings are not in the same plane, so the H_4_TCBP molecule presents a twisted tetrahedral configuration. Each H_4_TCBP molecule is connected to the carboxylic acid hydrogen bonds on the adjacent H_4_TCBP molecules through carboxylic acid hydrogen bonds on the four outer benzene rings, resulting in a robust 3D porous framework (Figure 1a). It has one-dimensional porous channel along the a-axis direction. Among them, the rhombohedral channed aperture is 17.81 × 26.34 Å. The solvent filling space of HOF-TCBP is calculated to be 56% of the total crystal volume. The BET surface area is 2066 m^2^g^−1^ [13]. The diffraction peaks of the experimentally synthesized samples are consistent with the PXRD patterns of the simulated structures (Figure 1b), indicating that the synthesized samples are pure phase. The morphology of HOF-TCBP was observed by scanning electron microscope (SEM) (Figure 1c,d), and the results showed that the synthesized samples were rod-shaped with the length of about several micrometers.

The HOF-TCBP solid powder shows photoluminescence (PL) characteristics at 392 nm, which is basically consistent with the fluorescence peak of organic construction molecule H4TCBP. The fluorescence characteristics of HOF-TCBP can be attributed to the fluorescence of organic construction molecule H4TCBP (Figure 2a). For HOF-TCBP solution dispersed in ethanol, it shows a strong fluorescence emission peak of 359 nm, which is about 33 nm blue shift compared with the emission peak of HOF-TCBP solid powder. This may be due to the fact that HOF-TCBP in liquid is relatively dispersed and the stacking effect is not obvious while HOF-TCBP in solid powder is densely stacked to form excimer, which leads to the red shifted PL compared with HOF-TCBP dispersed in ethanol. The pioneering work have illustrated the excimer formation phenomenon in several MOFs [47,48,49,50,51], it is noteworthy that this is the first work reporting the excimer formation in HOF materials.

The PL of different concentration of HOF-TCBP in ethanol was examined to illustrate the influence of solution concentration on the PL of this HOF (Figure 2b). When the concentration of HOF-TCBP suspension is 1.58 × 10^−3^ M, its emission peak is located at 369 nm. The PL emission blue-shifts with the dilution of the suspension concentration. When the HOF-TCBP suspension was gradually diluted, the emission peak blue-shifts to approximately 359 nm and basically remains unchanged. When the concentration is high, it is easy to form excimer, and the PL emission is red shift and the fluorescence intensity becomes weak. With the decrease of suspension concentration, the fluorescence intensity first increased with a small blue-shift of the fluorescence peak, which can be attributed to the excimer formation gradually diminished. When the concentration is 7.88 × 10^−4^ M, the fluorescence intensity reaches the strongest, and then the fluorescence intensity decreases, which can be attributed to the dilution of HOF-TCBP suspension.

Based on the above analysis, we selected the strong emission HOF-TCBP suspension with the concentration of 7.88 × 10^−4^ M to examine its potential application in metal ions sensing. The addition of a small amount of different metal ions to the HOF-TCBP ethanol solution lead to different degrees of change in the luminescence intensity. Analytes such as Ca^2+^, K^+^, Cd^2+^, Al^3+^ show no significant effect on the luminescence intensity of HOF-TCBP suspension, but the addition of Co^2+^, Cu^2+^, Zr^4+^ and Sc^3+^ ions quench the luminescence of HOF-TCBP suspension to varying degrees. In addition, there is a slight red shift of the emission peak after the addition of Zr^4+^ and Sc^3+^ ions (Appendix A). The red-shift is about 15 nm and 17 nm, respectively. The quenching effects of different metal ions on HOF-TCBP are Cr^3+^ > Fe^3+^ > Cu^2+^ > Sc^3+^ > Zr^4+^ > Co^2+^ (Appendix A). Especially when Cr^3+^ and Fe^3+^ ions were added, HOF-TCBP suspension showed a significant PL quenching effect (Figure 3a,c). When the concentrations of Cr^3+^ and Fe^3+^ were 2.44 × 10^−5^ M and 3.85 × 10^−5^ M respectively, the fluorescence intensity of HOF-TCBP could be quenched by more than 50%. When Cr^3+^ with a concentration of 9.09 × 10^−5^ M and Fe^3+^ with concentration of 1.03 × 10^−4^ M were added, the fluorescence of HOF-TCBP could be almost completely quenched. The HOF-TCBP material shows excellent sensing performance towards Cr^3+^ and Fe^3+^, and the detection limit reaches 0.689 μM and 2.516 μM (≈36 ppb and 141 ppb) (Appendix A), respectively, showing comparable or lower detection limits compared with previously reported MOF luminescent materials or other probes (Appendix A). The quenching efficiency of Cr^3+^ and Fe^3+^ ions can be quantitatively explained with the Stern Volmer equation I_0_/I = 1 + KSV [Q]. Where I_0_ is the fluorescence intensity of HOF-TCBP ethanol solution, I is the fluorescence intensity after adding Cr^3+^ or Fe^3+^ ions, and [Q] is the concentration of Cr^3+^ or Fe^3+^ ions. It is worth noting that the HOF-TCBP solution has a linear relationship with Cr^3+^ and Fe^3+^ ions in the low concentration range. As shown in Figure 3b,d, the ion quenching coefficients Ksv of Cr^3+^ and Fe^3+^ are about 8.50 × 104 M^−1^ and 3.01 × 104 M^−1^, respectively.

In addition, we also examined the sensing properties of HOF-TCBP towards different anions. When a small amount of Cr_2_O_7_^2−^ was added, the PL emission of HOF-TCBP was significantly quenched (Figure 4a) while there was no significant change with the addition of other anions, such as Ac^−^, Br^−^, Cl^−^, CO_3_^2−^, F^−^, HPO_4_^2−^, SiO_3_^2−^, SO_3_^2−^, and SO_4_^2−^ (Figure 4b). The fluorescence intensity of HOF-TCBP solution was linearly related to the amount of Cr_2_O_7_^2−^ added in the low concentration range. This relationship can be explained by the stern Volmer equation. The calculated quenching coefficient Ksv is about 3.62 × 10^4^ M^−1^ (Appendix A), and the detection limit is 1.638 μM (≈0.35 ppm) (Appendix A). It exhibits a superior detection limit over some previously reported luminescent materials (Appendix A). We further studied the selective detection of Cr_2_O_7_^2−^ in the presence of other anions and carried out the anti-interference experiments. The results showed that in the presence of other analytes, such as Ac^−^, Br^−^, Cl^−^, CO_3_^2−^, F^−^, HPO_4_^2−^, SiO_3_^2−^, SO_3_^2−^, SO_4_^2−^, etc., Cr_2_O_7_^2−^ still effectively quenched the fluorescence emission of HOF-TCBP. The decrease of fluorescence intensity indicates that HOF-TCBP also has a good ability to selectively detect Cr_2_O_7_^2−^ in the presence of other anions (Appendix A).

The addition of Cr^3+^, Fe^3+^, and Cr_2_O_7_^2−^ ions led to the fluorescence quenching of HOF-TCBP solution, while the XRD patterns of HOF-TCBP powder soaked with Cr^3+^, Fe^3+^, and Cr_2_O_7_^2−^ ions were consistent with the original pattern (Appendix A), indicating that the framework of HOF-TCBP does not collapse after the addition of Cr^3+^, Fe^3+^, and Cr_2_O_7_^2−^ ions. According to the FT-IR spectrum (Appendix A), the peak positions of HOF-TCBP before and after immersion of metal ions are consistent, indicating that there is no new chemical bond formation after the addition of the metal ions. Further study of a fluorescence lifetime can distinguish static quenching and dynamic quenching. When the analyte is added, the fluorescence lifetime is basically unchanged, which is static quenching; with fluorescence lifetime decay, the quenching process is considered to be dynamic. As shown in Appendix A, the fluorescence lifetime remains basically unchanged after adding the analyte (Cr^3+^, Fe^3+^, Cr_2_O_7_^2−^). This indicates that static quenching is dominant. We speculate that the mechanism of fluorescence quenching is fluorescence resonance energy transfer (FRET) and light induced electron transfer (PET). The absorption spectra of Fe^3+^ and Cr_2_O_7_^2−^ effectively overlap with the excitation and emission spectra of HOF-TCBP (Appendix A), indicating that the quenching mechanism of HOF-TCBP after adding Fe^3+^ and Cr_2_O_7_^2−^ may be attributed to competitive absorption and fluorescence resonance energy transfer. It is observed that the fluorescence quenching degree after adding Fe^3+^ and Cr_2_O_7_^2−^ is similar (Figure 3c and Figure 4a), while the effective overlap area between Cr_2_O_7_^2−^ absorption spectrum and HOF-TCBP emission spectrum is larger than that of Fe^3+^, so there may be other mechanisms for the quenching effect of Fe^3+^ to HOF-TCBP. The 3d orbit of Fe^3+^ is half full [52], which makes it easy to absorb electrons, thus electron transfer may also contributes to the fluorescence quenching for Fe^3+^. For Cr^3+^ ions, there is almost no spectral overlap, excluding the quenching mechanism of fluorescence resonance energy transfer and competitive absorption. Cr^3+^ show strong ability to gain an electron [53], thus the quenching mechanism of Cr^3+^ may be electron transfer.

Because HOFs materials show the characteristics of good biocompatibility and low cytotoxicity, we also explore the dopamine fluorescence detection application of HOF-TCBP materials. Dopamine (DA) is an important neurotransmitter, which plays an important physiological role in the function of nervous system [39,40]. Many related cognitive impairment diseases are related to the abnormal concentration of DA in the brain. Therefore, the research on the detection of DA is very important for the diagnosis and monitoring of the disease [54]. A series of dopamine fluorescence titration experiments were carried out in HOF-TCBP aqueous solution with pH = 2–6. The addition of a small amount of DA effectively quenched the fluorescence emission of HOF-TCBP. In different pH environments, HOF-TCBP shows similar quenching effect towards DA (Appendix A). The fluorescence sensing of DA is further studied in the environment of pH = 6. When adding 1.48 × 10^−4^ M DA to HOF-TCBP solution, the fluorescence intensity can be quenched to half of the original intensity; when 5.21 × 10^−4^ M DA was added, the fluorescence quenching degree reached almost 90% (Figure 5a). While the other analytes, such as cysteine, phenylalanine, glutamic acid, urea, glucose, tryptophan, and aspartic acid, show negligible fluorescence intensity to HOF-TCBP (Figure 5b). Therefore, HOF-TCBP shows excellent selectivity for DA. The detection limit (LOD) of HOF-TCBP to DA is 36.57 mM (Appendix A), and the quenching coefficient Ksv value is 9.78 × 104 M^−1^ (Appendix A). Interestingly, as shown in Appendix A, the fluorescence intensity changed with time. After adding DA, the fluorescence intensity was rapidly quenched within 10 s, and the fluorescence intensity remained basically unchanged after 40 s. Therefore, the response of HOF-TCBP to DA is very fast.

From the above knowledge, the fluorescence intensity of HOF-TCBP decreased after the addition of DA, while the XRD pattern of HOF-TCBP powder with a large amount of (1.0 × 10^−2^ M) DA was the same as the original (Appendix A), indicating that the backbone of HOF-TCBP does not collapse. In different acidic environments, the FT-IR spectra of HOF-TCBP immersed in DA solution were consistent with those of HOF-TCBP not immersed in DA solution, indicating that no new chemical bonds were formed (Appendix A). The fluorescence lifetime curve shows that the fluorescence lifetime of HOF-TCBP exhibits subtle decrease after adding DA (Appendix A). Therefore, the dynamic collision effect may exist during the quenching process. The UV-Vis absorption spectrum of DA shows a small spectrum overlap with emission spectra of HOF-TCBP, while the adsorption spectrum shows partially spectrum overlap with the excitation spectrum of HOF-TCBP (Appendix A), indicating that there may exist competitive absorption and fluorescence resonance energy transfer between DA and HOF-TCBP, which results from the fluorescence quenching of HOF-TCBP. The one-dimensional porous channels in the HOF-TCBP structure, and the high specific surface area increases the contact opportunities with target molecules such as DA, thereby improving the detection sensitivity and response speed.

In order to evaluate the feasibility of the sensing experiment in biological samples, target analytes were detected in real samples of fetal bovine serum. The fetal bovine serum was added with different concentrations of DA, and the content of DA in the actual samples was analyzed and determined by the standard addition method. As shown in Table 1, the recoveries of DA in fetal bovine serum samples ranged from 101.1% to 104.9% with acceptable RSD values, demonstrating that HOF-TCBP can be used for quantitative detection of DA in serum.

## 3. Experimental

### 3.1. Materials and Methods

All chemicals were obtained commercially and used without further processing and purification. Dopamine (98%), L-cysteine (99%), L-phenylalanine (99%), L-glutamic acid (99%), Urea (99%), D-glucose (99%), L-tryptophan (99%), L-aspartic acid (99%), and N, N-dimethylformamide (AR, 99.5%) were purchased from Shanghai Macklin Biochemical Co., Ltd. (Shanghai, China). The Powder X-ray diffraction (PXRD) patterns of HOF-TCBP were measured with an X-ray diffractometer (D2 PHASER) with Cu Kα radiation (λ = 1.54056 Å) at a scan rate of 5°/min and a scan range of 5°–50°. Field emission scanning electron microscopy (FE-SEM) images were obtained with a MODEL SU8010, Hitachi. The PL spectrum were obtained with an F4600 fluorescence spectrometer. FT-IR was carried out on a Nicolet iS50 spectrometer from Thermo Fisher Scientific (Waltham, MA, USA). UV-Vis absorption spectrum was recorded on Shimadzu UV-3600 spectrometer.

### 3.2. Synthesis

#### Synthesis of HOF-TCBP

The 3,3′,5,5′-tetrakis-(4-carboxyphenyl)-1,1′-biphenyl (H_4_TCBP) (100 mg, 0.158 mmol) and DMF (0.7 mL) were added to a 20 mL vial. After 10 min of sonication, it was allowed to stand for several days, and the liquid slowly evaporated and crystallized. Then, 0.5 mL of DMF were added to dissolve it, and 5 mL of acetone was added for anti-dissolving, mixed well, and filtered. Finally, it was washed several times with acetone and air-dried to obtain a white powder.

### 3.3. Metal Ions Sensing

In metal ion sensing experiments, a 10 mg HOF-TCBP powder sample was weighed then dispersed into 200 mL of an ethanol solvent, and after sonication for 30 min, a HOF-TCBP suspension was obtained. A series of nitrate solutions A(NO_3_)_x_ (10^−3^M, A = Ca^2+^, K^+^, Cd^2+^, Al^3+^, Co^2+^, Zr^4+^, Sc^3+^, Cu^2+^, Fe^3+^, Cr^3+^) was prepared according to the titration method, 2 mL of the HOF-TCBP suspension was gradually added, and then the fluorescence spectrum (excitation wavelength: 304 nm, monitoring wavelength: 359 nm) was measured.

### 3.4. Anion Sensing

For the anion sensing experiments, the sodium salt aqueous solutions of Ac^−^, Br^−^, Cl^−^, CO_3_^2−^, F^−^, HPO_4_^2−^, SiO_3_^2−^, SO_3_^2−^, SO_4_^2−^, and Cr_2_O_7_^2−^ with a concentration of 10^−3^ M were first prepared, and then different amounts of the above stock solutions were added to 2 mL HOF-TCBP suspension, respectively; the luminescence data was collected after standing overnight. In the anti-interference experiment of Cr_2_O_7_^2−^, 230 μL of another nine anion solutions were added to 2 mL HOF-TCBP suspension, and then the Cr_2_O_7_^2−^ solution was added incrementally, and the luminescence data was collected after standing overnight (excitation wavelength: 304 nm, monitoring wavelength: 359 nm).

### 3.5. Dopamine Sensing

A DA stock solution was prepared at a concentration of 10 mM. Meanwhile, the HOF-TCBP samples were dispersed in a phosphate buffer solution (5 mg/100 mL) with pH 2–6, respectively, and ultrasonically treated for 30 min. Then, 2 mL HOF-TCBP suspensions with different pH were taken respectively, and the DA solution was added dropwise in order to measure the fluorescence spectrum (excitation wavelength: 330 nm, monitoring wavelength: 397 nm). In addition, seven solutions were also prepared at a concentration of 10 mM, cysteine (Cys), phenylalanine (Phe), glutamic acid (Glu), urea (Urea), glucose (Glc), tryptophan (Trp), and aspartic acid (Asp) were added to 2 mL HOF-TCBP suspension at pH = 6, respectively, and the fluorescence was measured for selectivity experiments. The fetal bovine serum was diluted 30-fold before the measurement of the real sample fetal bovine serum.

## 4. Conclusions

In summary, a porous HOF-TCBP was synthesized using a solvent diffusion method. Interestingly, fluorescence detection experiments show that HOF-TCBP can be used as a multi-response luminescence sensor for the sensitive detection of metal ions Fe^3+^, Cr^3+^, and anion Cr_2_O_7_^2−^, respectively, and HOF-TCBP maintains stability in a pH = 2–6 buffer solution, and selectively detects biomolecular DA through the shut-off effect. At the same time, the real sample fetal bovine serum was also verified, and the feasibility of the sensing experiment in biological samples was also verified. In addition, the characterization by PXRD, SEM, UV-vis, and FT-IR was carried out to investigate the mechanism of the interaction between HOF-TCBP and analytes. The quenching mechanism of HOF-TCBP by Fe^3+^ and Cr_2_O_7_^2−^ is mainly attributed to competitive absorption and fluorescence resonance energy transfer. In addition, electron transfer is also a reason for the quenching of HOF-TCBP by Fe^3+^ and Cr^3+^. For DA, the quenching mechanism is mainly attributed to competitive absorption and fluorescence resonance energy transfer. This work further investigates the luminescence properties and sensing behaviors based on HOFs.

## Figures and Tables

**Figure 1 molecules-27-08750-f001:**
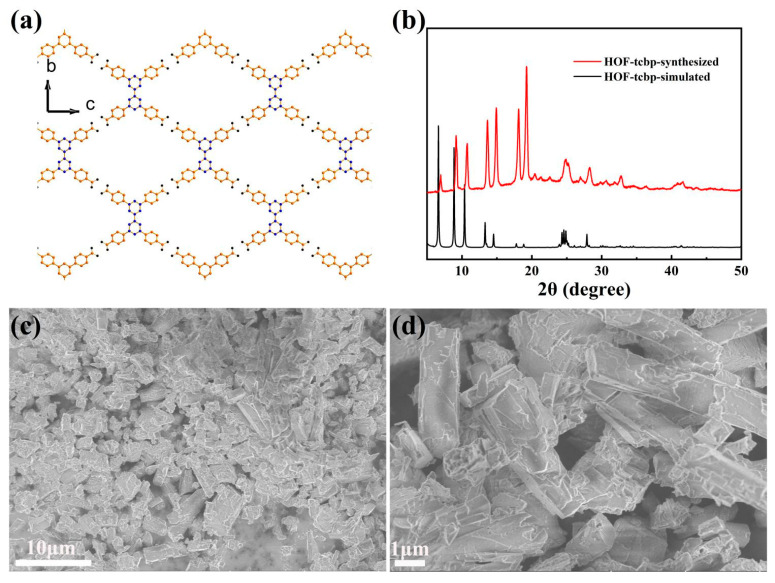
(**a**) HOF-TCBP framework linked by organic building blocks H_4_TCBP through hydrogen bonding; (**b**) the PXRD patterns of experimentally synthesized HOF-TCBP (red curve) and simulated data (black curve); (**c**,**d**) the SEM images of HOF-TCBP.

**Figure 2 molecules-27-08750-f002:**
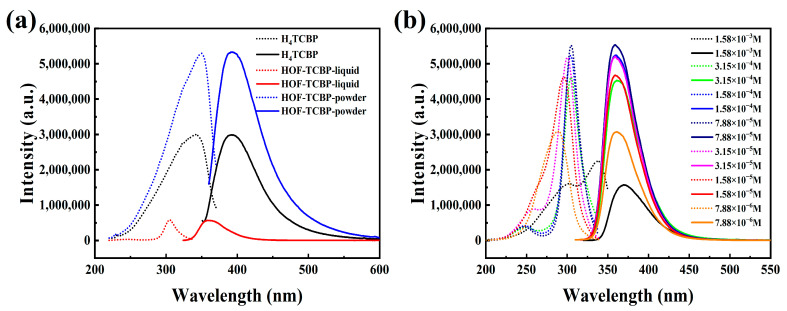
(**a**) Excitation (short point line) and emission spectra (solid line) of organic building molecules H_4_TCBP, HOF-TCBP powder and HOF-TCBP ethanol solution; (**b**) fluorescence spectra of HOF-TCBP ethanol solutions at different concentrations.

**Figure 3 molecules-27-08750-f003:**
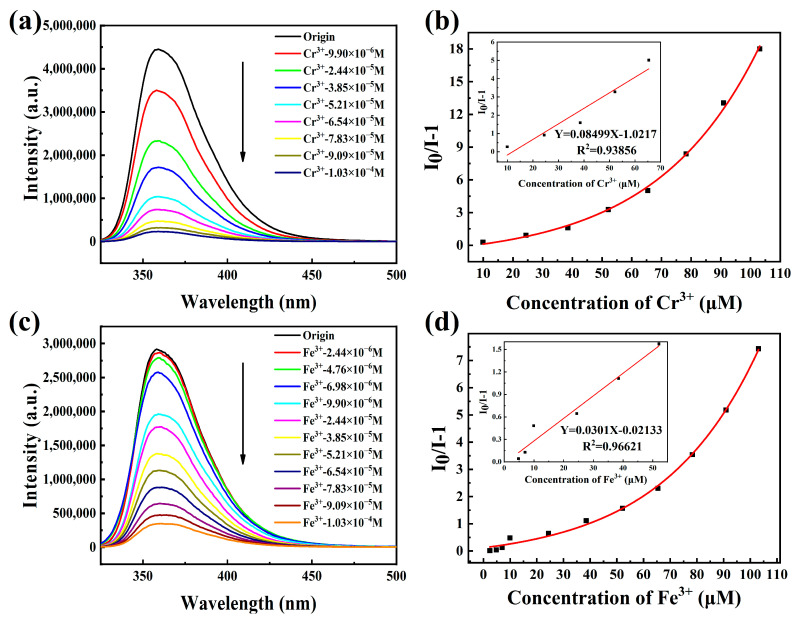
(**a**,**c**) The PL spectra of HOF-TCBP with different Cr^3+^ and Fe^3+^ ion concentrations added (excitation wavelength: 304 nm, monitoring wavelength: 359 nm); (**b**,**d**) S–V plots of Cr^3+^ and Fe^3+^; inset: the above figures are the linear fitting curves at low concentrations, respectively.

**Figure 4 molecules-27-08750-f004:**
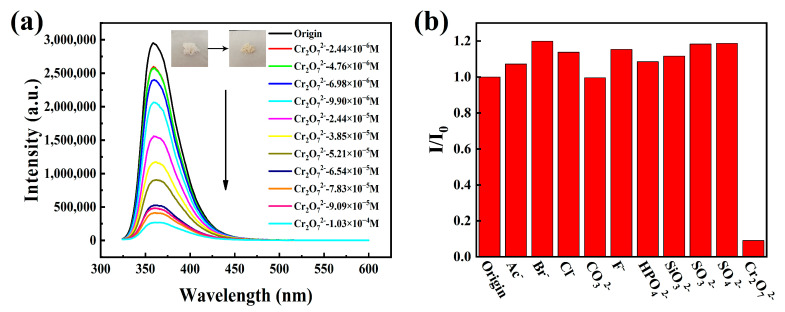
(**a**) Fluorescence spectra of HOF-TCBP after adding different concentrations of Cr_2_O_7_^2−^; inset: the above picture shows powder HOF-TCBP without Cr_2_O_7_^2−^ and with 10^−3^ M Cr_2_O_7_^2−^; (**b**) the luminescence intensity of Cr_2_O_7_^2−^ and other interfering substances at 359 nm at the same concentration (1.0 × 10^−3^ M).

**Figure 5 molecules-27-08750-f005:**
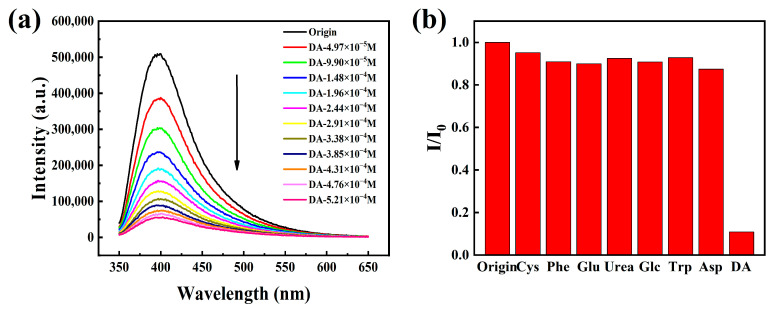
(**a**) The PL spectra of HOF-TCBP with different DA contents in pH = 6 buffer solution (excitation wavelength: 330 nm, monitoring wavelength: 397 nm); (**b**) the emission intensity of HOF-TCBP of 1.0 × 10^−2^ M different organic compounds at pH = 6 buffer solution at 397 nm.

**Table 1 molecules-27-08750-t001:** Detection of DA in fetal bovine serum.

No.	Spiked (µM)	Measured (µM)	Recovery (%)	R.S.D. (n = 3, %)
1	50	51.4	102.8	2.62
2	100	104.9	104.9	2.26
3	150	151.7	101.1	1.54

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
