# Peer review of "Multi-Responsive Sensor Based on Porous Hydrogen-Bonded Organic Frameworks for Selective Sensing of Ions and Dopamine Molecules"

_molecules, 2022, doi:10.3390/molecules27248750_

Round 1

Reviewer 1 Report

1.     All the figures should be improved, especially for Fig.2 and 3.

2. The manuscript contains spelling/grammatical errors. So, the language should be polished thoroughly.

3. Source and purity of all chemicals used should be specified in the experimental section 4. The structural stability should be confirmed by the morphology of samples after multiple sensing tests.

5. For introduction, some updated refs should be cited, such as Micropor. Mesopor. Mat, 341(2022) 112098; Inorganics 2022, 10, 202; Polyhedron 2019,157, 420–427; Dalton Trans.,2020,49, 4741-4750; New J. Chem., 2022, 46, 19577–19592 and CrystEngComm, 2022, 24, 7157–7165.

6. The conclusion section should be give more informative values.

7. For sensing the metal ions, the sensing process of the interfering substances should be done.

8. The recycle test should be checked and added.

9. Give a sensing mechanism for the full work.

Author Response

1. All the figures should be improved, especially for Fig.2 and 3. Response: Improvements have been made in Figure 2 and Figure 3. 2. The manuscript contains spelling/grammatical errors. So, the language should be polished thoroughly. Response: The language has been checked and polished thoroughly. 3. Source and purity of all chemicals used should be specified in the experimental section 4. The structural stability should be confirmed by the morphology of samples after multiple sensing tests. Response: The source and purity of the chemicals used were added in the experimental section. 5. For introduction, some updated refs should be cited, such as Micropor. Mesopor. Mat, 341(2022) 112098; Inorganics 2022, 10, 202; Polyhedron 2019,157, 420–427; Dalton Trans.,2020,49, 4741-4750; New J. Chem., 2022, 46, 19577–19592 and CrystEngComm, 2022, 24, 7157–7165. Response: The above references have been added in the references. 6. The conclusion section should be give more informative values. Response: More descriptions of the conclusions are added in the conclusion section on page 15. 7. For sensing the metal ions, the sensing process of the interfering substances should be done. Response: The sensing process of the interfering substances is in the support information (Figure S1-S9) and line 170-175 in page 9 in the manuscript. 8. The recycle test should be checked and added. Response: For the HOF suspension in this work, it is hard to carry out the recycle test, however further recycle test will be checked on the HOF-based membranes in the future. 9. Give a sensing mechanism for the full work. Response: The mechanism of HOF-TCBP quenching was analyzed in the manuscript, line 176-198, page 10. The summary of the quenching mechanism was added in the conclusion part of page 15.

Reviewer 2 Report

In this paper, the author synthesized porous material HOF-TCBP by self-assembly, which showed good sensitivity and selectivity to metal ions Fe3+, Cr3+ and anion Cr2O72-. In addition, the HOF-based material was for the first time applied to the sensing of neurotransmitter dopamine, showing fast response, sensitivity and anti-interference to many interferences, and was also applied in actual biological samples. These results indicate that the multi-response sensor based on HOF-TCBP has great potential in practical applications. However, some minor changes are needed before publication. Details are as follows.

1. The porous information (eg. The pore size and pore structure) was not given in the manuscript. Please describe these important structure information of HOF-TCBP in detail.

2. Is the HOF-TCBP material stable after sensing the analyte? Please provide the PXRD pattern after the HOF sensing.

3. Please check the superscripts and subscripts of the ions involved in the manuscript.

Author Response

  1. The porous information (eg. The pore size and pore structure) was not given in the manuscript. Please describe these important structure information of HOF-TCBP in detail.

Response: On page 5 of the manuscript, more descriptions of the pores are added in line 1.

  1. Is the HOF-TCBP material stable after sensing the analyte? Please provide the PXRD pattern after the HOF sensing.

Response: In the support information diagrams S23 and S35, the PXRD was measured after the HOF sensing. The peak position does not change much compared with the synthetic sample, indicating that HOF-TCBP remains stable (line 176-178 in page 9 in the manuscript).

  1. Please check the superscripts and subscripts of the ions involved in the manuscript.

Response: The superscript and subscript of ions in the manuscript have been checked and corrected.